# Sleep Bruxism Contributes to Motor Activity Increase during Sleep in Apneic and Nonapneic Patients—A Polysomnographic Study

**DOI:** 10.3390/biomedicines10102666

**Published:** 2022-10-21

**Authors:** Tomasz Wieczorek, Monika Michałek-Zrąbkowska, Mieszko Więckiewicz, Grzegorz Mazur, Joanna Rymaszewska, Joanna Smardz, Anna Wojakowska, Helena Martynowicz

**Affiliations:** 1Department and Clinic of Psychiatry, Wroclaw Medical University, 50-367 Wroclaw, Poland; 2Department and Clinic of Internal Medicine, Occupational Diseases, Hypertension and Clinical Oncology, Wroclaw Medical University, 50-367 Wroclaw, Poland; 3Department of Experimental Dentistry, Wroclaw Medical University, 50-367 Wroclaw, Poland

**Keywords:** motor activity, sleep bruxism, sleep apnea, obstructive

## Abstract

Background: Jaw motor activity (MA) in sleep bruxism (SB) has been demonstrated to accompany lower limb movements. However, it remains unknown whether SB activity coexists with other types of movements and what the possible underlying mechanisms of such temporal coexistence are. In obstructive sleep apnea (OSA), increased movement activity is also reported, including SB activity; however, no studies have compared MA in apneic and nonapneic SB patients. Aim: This cross-sectional study focused on the phenomenon of “big body movements” in patients with either SB or OSA (or both) and intended to identify the primary factors contributing to their appearance, using polysomnography (PSG) recording. Methods: A whole-night videoPSG was carried out in 287 participants, and 124 apneic and 146 nonapneic participants were selected for the study. In both groups, participants were further divided into no SB, moderate SB, and severe SB (SSB) subgroups based on their bruxism episode index (BEI). MA was recorded using a built-in sensor of the central PSG unit located on the participant’s chest during the examination. Results: The presence of SB was related to the higher intensity of MA in both apneic and nonapneic participants, though in general the MA level was higher in apneic participants, with the highest level observed in SSB apneic participants. Conclusions: SB might contribute to MA. The prevalence of SB might be higher in nonapneic patients due to phasic and mixed SB activity, whereas the SB phenotype seems to be less relevant in apneic patients. SB activity is likely to increase MA in non-REM 1 sleep.

## 1. Introduction

Sleep bruxism (SB) is defined as “a masticatory muscle activity during sleep that is characterized as rhythmic (phasic) or nonrhythmic (tonic) and is not a movement disorder or a sleep disorder in otherwise healthy individuals” [1]. It is a phenomenon that can be perceived as either a sleep-related motor behavior, a risk factor, or a protective factor [1,2]. It affects around 13% of the general population, more commonly younger adults, with its prevalence decreasing over age [1,2,3,4]. It might contribute to significant clinical problems such as teeth damage, pain in masticatory muscles, fatigue of masticatory muscles, damage to temporomandibular joints (TMJ), headaches, and failures in prosthodontic treatments [1,2,3,4]. In recent years, SB has become an intensively studied topic, and several significant conclusions have emerged. One of these findings is that the pain intensity seems to be not dependent on SB polysomnography (PSG) parameters [5]. Considering pain as one of the most problematic consequences of SB, Ferrillo et al. have reported, based on a conducted meta-analysis, that conservative approaches including rehabilitative therapies (like splint appliance and laser therapy) might be effective in the treatment of pain in TMJ disorders [6]. New treatment methods are emerging, among them regeneration of TMJ with the use of mesenchymal stem cells [7]. In the global novel problem of the COVID-19 pandemic, even some modern teledentistry approaches have been suggested as applicable in the treatment of TMJ disorders [8].

In recent years, the relationship between SB and sleep-related breathing disorders, especially obstructive sleep apnea (OSA), has been widely discussed in the medical literature. OSA is a respiratory sleep disorder characterized by common partial (hypopnea) or complete (apnea) obstruction of the upper airways during sleep, which leads to desaturation, arousals, sleep fragmentation, and increased sympathetic activity [9]. Tan et al. reported that around 33–50% of OSA patients also suffer from SB [10]. A recent study reported that SB might be independently contributing to snoring, regardless of the body position; however, the supine position is more likely to increase both SB and snoring severity [11]. Other significant comorbidities of OSA include insomnia, periodic limb movement during sleep (PLMS) [12], and parasomnias [13]. OSA is also an individual risk factor for cardiovascular diseases such as hypertension, strokes, atrial fibrillation, and coronary artery disease [14].

Both hypopneic or apneic episodes and SB episodes have been demonstrated to increase sympathetic activity during sleep [15]. The question that arises is whether similar autonomic hyperstimulation appearing in SB also leads to increased body movement activity, and so far, mostly studies with a small sample size were published, suggesting that both SB and other short body movements might share a similar underlying central mechanism [16]. In a recent study, Shiraishi et al. reported that, in children, more than 70% of rhythmic masticatory muscle activity episodes were associated temporarily with cortical and motor arousals [17].

The literature review shows that older studies, i.e., those published in the 1980s and 1990s, suggested that body movements are related to sleep stage transition, may increase shortly before and after rapid eye movement (REM) sleep, and usually decrease before slow wave sleep [18]. It was also observed that the frequency and duration of body movements usually remain similar during subsequent nights [19]. However, these studies were not able to indicate the exact underlying mechanisms of their observations, though they highlighted that body movements are not mere artifacts and these phenomena need to be studied.

Since then, many terms have been used to describe body movements. In this study, the primary focus was the activity of different body parts, including upper and lower extremities, trunk, and head/neck, which has not yet been clearly defined. Some propositions of proper nomenclature were available, such as gross body movements [20,21]. Fukumoto et al. classified body movements into gross movements (GM) (including two or more limbs and the trunk, lasting longer than 0.5 s), localized movements (just one limb movement, lasting longer than 0.5 s), and twitch movements (one limb or other parts of the body, lasting shorter than 0.5 s); however, this classification was made based on movements in infants [21]. Recently, a new term, “large muscle movements” (LMMs), was used as a part of the diagnostic criteria for a new pediatric sleep disorder called restless sleep disorder [22]. LMMs were defined as “gross, visible body movements, involving large muscle groups of the whole body, all four limbs, arms, legs, or head. The sleep movements may be characterized further as frequent repositioning, disruption of the bedsheets, or falling out of bed” [22]. Besides different terminologies, new methods of movement measurement were introduced, e.g., extraction of artifacts from electrooculograms and electromyograms from PSG recording [23,24], video recording analysis [20,25,26], actigraphy [27,28,29], and heat sensors located under the mattress [30]. Some of these methods were also proposed as a possible means to carry out sleep stage scoring based on motor activity (MA) [25,30].

These days, many sleep-related movement disorders such as SB and PLMS can be distinguished, but not all body movement phenotypes can be easily classified. This topic remains significant because many studies suggested that SB and PLMS also contribute to increased cardiovascular risk [15,31]. Some previous studies suggested that body movements in sleep may decrease with age, but in the elderly, these movements are more likely to result in sleep stage transition or awakening [23,24]. Changes in MA may indicate other sleep (insomnia) or even psychiatric disorders (depression), and are a basic phenomenon underlying the whole concept of actigraphy diagnostics [28,29].

This cross-sectional study focused on the phenomenon of “big body movements” in patients with either SB or OSA (or both) and intended to identify the primary factors contributing to their appearance, based on PSG recording with a built-in acceleration sensor in the central unit of the PSG device. A direct comparison of apneic and nonapneic patients may bring a new insight into the phenomenon of MA in sleep and its relation to sleep stages, arousals and bruxism activity.

## 2. Materials and Methods

### 2.1. Participants

The participants of this study were patients hospitalized in the Department and Clinic of Internal Diseases, Occupational Diseases, Hypertension, and Clinical Oncology at Wroclaw Medical University. This study was approved by the Ethical Committee of Wroclaw Medical University (ID KB-195/2017 issued on 20 April 2017) and was conducted in accordance with the declaration of Helsinki. All patients signed an informed consent form for participating in this study. Information regarding clinical trial registration is available at www.ClinicalTrials.gov (accessed on 29 September 2022) (identifier NCT03083405). Patients who met the following criteria were included in this study: adult age, clinical SB suspicion (assessed by an experienced dentist), and willingness to participate in the study. Patients were excluded if they had a history of neurological, degenerative, severe mental or cardiovascular disorders, the presence of cancer, were pregnant, or were taking medications that affect the neuromuscular system.

### 2.2. Polysomnographic Evaluation

All patients underwent overnight videoPSG using Nox-A1 (Nox Medical, Reykjavik, Iceland) in the Sleep Laboratory of the Department and Clinic of Internal Medicine, Occupational Diseases, Hypertension, and Clinical Oncology at Wroclaw Medical University, Poland. Based on the standard criteria for sleep scoring recommended by the American Academy of Sleep Medicine (AASM), a manual assessment of the registered data was performed on a 30 s epoch basis. The following registered parameters were included: total sleep time (TST), sleep latency (SL), REM latency (REML), sleep efficiency, duration of wake episodes after sleep onset, arousal index, percentages of non-REM sleep stages 1–3, and percentage of REM sleep. Abnormal respiratory events were scored according to the standard criteria of the AASM Task Force (Berry et al., 2018), and the apnea–hypopnea index (AHI) was calculated as follows. Apnea is defined as the absence of airflow for ≥10 s, and hypopnea is defined as the reduction in the amplitude of breathing by ≥30% for ≥10 s with a ≥3% decline in the blood oxygen saturation level, leading to arousal from sleep. SB was assessed using bilateral masseter electromyography with parallel audio and video evaluation. SB episodes were scored into three phenotypes according to the AASM standards: phasic, tonic, and mixed. The bruxism episode index (BEI) was evaluated for each participant in general and separately for all sleep stages and for different phenotypes. In addition, the nontonic BEI was calculated as the sum of the phasic BEI and the mixed BEI.

Activity parameters were measured along with whole-night PSG recording, using the built-in sensor of the central unit of the mobile PSG device (Nox A1). It records 3-axis gravity signals with a range of −1 to 1 times of Earth’s gravity units (g). The axes were positioned in such a way that the x-axis goes through the chest, the y-axis goes from shoulder to shoulder, and the z-axis goes from the feet to the head. From these signals, the angle of rotation was measured from the arc tangent of y and x. The acceleration signal was used to derive the position signal measured in a 360°scale and the activity signal measured in g/s. The device supports a 20 Hz sampling rate of the activity signals with a noise level of <20 mGRMS on each axis. The activity signal was calculated from raw gravity signals measured using an accelerometer in the recording device, where it was the absolute change of the length of the x- and y-axis vectors. The z-axis, which runs parallel to the patient’s body, did not contribute relevant information and was therefore omitted from the calculation. An example of PSG recording with a marked MA episode is presented on Figure 1. The following activity parameters were recorded and calculated:

Activity duration (AD), total time spent in activity during sleep;

Percentage of AD in TST, i.e., the percentage of whole TST spent in activity;

AD in apnea/hypopnea, the total time of activity during apnea or hypopnea episodes; episodes occurring within the apnea/hypopnea episode and/or 3 s before or after were counted;

AD in bruxism, the total time of activity during bruxism episodes; episodes occurring within the bruxism episode and/or 3 s before or after were counted;

Percentage of AD in bruxism of total AD, i.e., the percentage of total AD spent on activity in bruxism episodes;

AD in desaturation, the total time of activity during desaturation (≥3% of SpO_2_) episodes, episodes occurring within the desaturation episode and/or 3 s before or after were counted;

AD in N1, the total time of activity spent in N1 sleep;

AD in N2, the total time of activity spent in N2 sleep;

AD in N3, the total time of activity spent in N3 sleep;

AD in REM, the total time of activity spent in REM sleep;

Activity count (AC) in TST, the number of activity episodes spent in activity during 1 h of sleep;

AC in apnea/hypopnea, the number of activity episodes/h during apnea or hypopnea episodes; episodes occurring within the apnea/hypopnea episode and/or 3 s before or after were counted;

AC in bruxism, the number of activity episodes/h during bruxism episodes; episodes occurring within the bruxism episode and/or 3 s before or after were counted;

AC in desaturation, the number of activity episodes/h during desaturation (≥3% of SpO_2_) episodes; episodes occurring within the desaturation episode and/or 3 s before or after were counted;

AC in N1, the number of activity episodes spent in 1 h of N1 sleep;

AC in N2, the number of activity episodes spent in 1 h of N2 sleep;

AC in N3, the number of activity episodes spent in 1 h of N3 sleep;

AC in REM, the number of activity episodes spent in 1 h of REM sleep.

### 2.3. Statistical Analysis

Statistical analysis was carried out using the “Statistica” software, v. 13.3 (StatSoft, Cracow, Poland). The Shapiro–Wilk test was conducted to evaluate the normal distribution of the data. The Mann–Whitney *U* test was conducted for the nonparametric data to test the significance of differences between groups in the case of independent variables. Dependent variables were compared directly using the Wilcoxon’s matched pairs signed rank test. Subgroups were analyzed based on the Kruskal–Wallis Analysis of Variance (ANOVA) for the nonparametric data. Bonferroni correction was applied in case of multiple comparisons (Kruskal–Wallis ANOVA). Chi-square tests were used to verify the differences in qualitative variables. Correlation analysis was carried out using the Spearman’s correlation rank test. Statistical significance was set at *p* < 0.05 for all statistical methods used.

## 3. Results

A total of 287 participants were subjected to the inclusion and exclusion criteria mentioned earlier. Finally, 270 participants were included in the analysis (mean age = 42.15 ± 14.69; 157 female and 113 male); seventeen participants were excluded due to missing data (mostly being a consequence of technical issues during the PSG registration). The participants were classified into two main subgroups—those with confirmed obstructive apnea syndrome (124 participants, 44 female and 80 male; mean age = 51.16 ± 14.81) and those with excluded obstructive apnea syndrome (146 participants; 112 female and 34 male; mean age = 34.57 ± 10.37). The criterion of AHI > 5 was used to diagnose obstructive apnea syndrome of at least mild severity.

### 3.1. Non-OSA Group

According to SB severity, the non-OSA group was further classified into three subgroups following the BEI criterion: BEI < 2—no significant SB (NSB); BEI ≥ 2 and <4—moderate SB (MSB); and BEI ≥ 4—severe SB (SSB). SB was excluded in 47 participants (39 female and 8 male), and this group was labeled NSB. A total of 45 participants (38 female and 7 male) were present in the MSB group and 54 participants (36 female and 18 male) in the SSB group. The Kruskal–Wallis nonparametric ANOVA revealed no significant difference between the subgroups in terms of age (*p* = 0.055), and chi-square tests revealed no difference in the structure of the subgroups in terms of sex (*p* = 0.059).

Details regarding the age and PSG results for the whole non-OSA group and all three subgroups (NSB, MSB, and SSB) are presented in Table 1.

#### 3.1.1. Non-Parametric Direct Comparisons of Activity Parameters in Groups

The nonparametric data were compared using Kruskal–Wallis ANOVA and Mann–Whitney *U* tests (NSB, MSB, and SSB groups were compared directly). ANOVA revealed significant differences between the subgroups for all activity parameters except for AC in N3 (*p* = 0.10) and AD in N3 (*p* = 0.21). Detailed direct comparisons of the subgroups revealed further differences. AD, AD in bruxism, AD in N1, AC in TST, and AC in bruxism were different between all subgroups. In the direct comparison of the NSB and SSB groups, significant differences were observed for all parameters, except for AD in N3, which was similar in both groups (*p* = 0.11). Further details are presented in Table 2.

AD in apnea/hypopnea and AD in desaturation were compared with AD in bruxism in the whole group and all subgroups separately. Significant differences were observed between the whole group and the MSB and SSB subgroups (*p* < 0.001 in all cases), while no difference was observed between the whole group and the NSB subgroup (*p* = 0.19 for AD in apnea/hypopnea and *p* = 0.41 for AD in desaturation). Similar comparisons were made for AC in apnea/hypopnea and AC in desaturation with AC in bruxism. In the NSB subgroup, no differences were observed for AC in apnea/hypopnea (*p* = 0.47) and for AC in desaturation and AC in bruxism (*p* = 0.03). Significant differences were observed between the whole group and the MSB and SSB subgroups (*p* < 0.001 in all cases).

#### 3.1.2. Correlations

Spearman correlation coefficients were calculated for BEI (including different SB phenotypes) and various activity parameters measured during PSG. In the non-OSA group, moderate positive correlations of BEI were observed for AD, percentage of AD of TST, AD in N1, and AC in TST. Strong correlations were observed for AD in bruxism and AC in bruxism. Similar correlations were observed for phasic BEI and nontonic BEI (phasic BEI + mixed BEI), though Spearman *R* values were slightly lower in phasic BEI compared with total BEI and slightly higher in nontonic BEI compared with total BEI. On the contrary, a moderate correlation of tonic BEI was observed only with the AC in bruxism value; in the case of other variables, the correlation coefficient values were much lower, compared with BEI, phasic BEI, and nontonic BEI, with no other moderate or severe correlations. Detail are resented in Table 3. 

### 3.2. OSA Group

Based on SB severity, the OSA group was further divided into three subgroups using the same criteria as the non-OSA group. SB was excluded in 46 participants (18 female and 28 male), which was labeled the NSB group. A total of 24 participants (10 female and 14 male) were confirmed in the MSB group and 54 participants in the SSB group (16 female and 38 male). Kruskal–Wallis nonparametric ANOVA revealed significant differences between the subgroups in terms of age (*p* = 0.007), with the NSB group being significantly younger than other groups, and chi-square tests revealed no difference in the structure of the subgroups in terms of sex (*p* = 0.48).

Further details regarding age and PSG results for the whole OSA group and all three subgroups (NSB, MSB, and SSB) are presented in Table 4.

#### 3.2.1. Nonparametric Direct Comparisons of Activity Parameters in Groups

The nonparametric data were compared using Kruskal–Wallis ANOVA and Mann–Whitney *U* tests (NSB, MSB, and SSB groups were compared directly). ANOVA revealed significant differences between the subgroups in the case of AD in bruxism (*p* < 0.001), AD% in bruxism of total AD (*p* < 0.001), AD in N1 (*p* = 0.014), AC in bruxism (*p* < 0.001), AC in N1 (*p* < 0.001), and AC in TST (*p* = 0.025), although the last result (AC in TST) did not remain significant after the appliance of the Bonferroni correction (*p* > 0.017). Detailed direct comparisons of the subgroups revealed further differences. In all subgroups, only AD in bruxism, AD% in bruxism of total AD, and AC in bruxism were different. In the direct comparison of theNSB and SSB groups, significant differences were observed in AD, AD% of TST, AD in bruxism, AD% in bruxism of total AD, AD in N1, AC in bruxism, AC in N1, and AC in TST. Further details are presented in Table 5.

AD in apnea/hypopnea and AD in desaturation were compared with AD in bruxism in the whole group and all subgroups separately. Significant differences were observed in the whole group and all three subgroups (*p* < 0.001 for all parameters except for the MSB subgroup with *p* = 0.025 for AD in apnea/hypopnea and *p* = 0.015 for AD in desaturation). Similar comparisons were made for AC in apnea/hypopnea and AC in desaturation with AC in bruxism. In the whole group and the NSB and MSB subgroups, significant differences were observed (*p* < 0.001 in all cases except for the MSB subgroup with *p* = 0.03 for AC in apnea/hypopnea and *p* = 0.01 for AC in desaturation). In the SSB subgroup, no difference was observed for AC in apnea/hypopnea (*p* = 0.053), whereas a significant difference was observed for AC in desaturation (*p* = 0.032).

#### 3.2.2. Correlations

Spearman correlation coefficients were calculated for BEI (including different SB phenotypes) and various activity parameters measured during PSG. In the OSA group, strong correlations were observed for AD in bruxism and AC in bruxism. Similar correlations were observed for phasic BEI and nontonic BEI (phasic BEI + mixed BEI), though Spearman *R* values were slightly lower in the case of phasic BEI and nontonic BEI compared with total BEI. On the contrary, a moderate correlation of tonic BEI was observed only with AD in bruxism and AC in bruxism; in the case of other variables, correlation coefficient values were much lower compared with BEI, phasic BEI, and nontonic BEI, with no other moderate or severe correlations. Further details are presented in Table 6.

### 3.3. Non-OSA Group vs. OSA Group

Non-OSA and OSA groups (including NSB, MSB, and SSB subgroups) were compared directly using the Mann–Whitney *U* test. Significant differences were observed for almost all measured activity variables (except for AC in Bruxism and AC in N1) and age, whereas no significant differences were observed for bruxism activity variables (except for BEI in N1). Similar observations were made for subgroups; however, AD in bruxism differed significantly only in the SSB subgroup, whereas AD in N1 and AD in REM differed only in the NSB subgroups. In terms of SB parameters, tonic activity, and BEI in N1, N2 and REM were different between both SSB subgroups. In the MSB subgroups, only BEI in N1 differed significantly, and in the NSB subgroups, BEI in N3 reached statistical significance. Further details are presented in Table 7. AD and AC in TST data in all six subgroups are presented in Figure 2 and Figure 3, respectively.

As it is presented in Table 7, significant differences of age were observed in the case of all compared groups and subgroups. Additional Spearman correlation analysis was performed to check possible associations of age, AHI, BEI and MA parameters. In the OSA group (including MSB and SSB subgroups), moderate correlations were found with AHI. Age also correlated positively with a moderate strength with the AD in apnoea/hypopnea and desaturation episodes and AC in apnoea/hypopnea and desaturation episodes, but only in OSA, MSB and SSB subgroups. In the whole OSA group, the mentioned correlations remained significant, but on a weaker level. Age weakly correlated in a negative manner with other MA parameters and BEI (including AC and AD in TST and across different sleep stages), but these coefficients did not reach high values. These effects were marked stronger in the Non-OSA group and subgroups. Further details are presented in Table 8.

## 4. Discussion

This study used a relatively novel approach to address the question of MA measurement, using a built-in sensor of the central PSG unit, located on the patient’s chest throughout the examination. Technical details were described previously in the Materials and Methods section. From the practical point of view, trunk movements appearing together with (less often without) upper limb or neck/head movements were scored using this movement registration. Exclusive head/neck or limb (upper or lower) movements were also recorded; however, they required a relatively big amplitude to be recorded if no simultaneous trunk movement was observed. This indicates that movements partially meeting the criteria of GM or LMM were recorded in this study. As described in the Introduction section, GM included two or more limbs and the trunk [20,21]—in our study, possibly more movements were recorded than just GM—such as exclusive movements of the trunk and/or just one limb or the head/neck. When compared with LMM, the movements recorded in this study were more specific—as PLM was not recorded specifically, probably only relatively big leg movements not appearing together with trunk movements could have been recorded; so not every type of LMM was likely to be recorded using the present methodology [22]. In addition, it is important to note that both terms—GM and LMM—in the mentioned context were used to describe movements in infants or children. In the case of activity recorded in the present study, the term “MA” was used to name the type of activity. Future studies with proper classification of movement types in sleep are highly needed, including proper phenotyping and specific morphology/timing criteria.

One of the most important observations in this study is that in both primary groups (OSA and non-OSA), two basic activity parameters (AD and AC in TST) were lowest in the NSB groups, higher in the MSB groups, and highest in the SSB groups. This might show that SB increases activity parameters regardless of the presence of OSA. Based on the fact that many types of movement activity (such as SB and PLMS) often follow microarousals and sympathetic activation [17,32,33,34,35,36,37], the observation that activity parameters were generally higher in the OSA group can be explained, as hypopnea and apnea episodes often result in arousals and sympathetic activation [9,38]. Kato et al. suggested that different types of movements are more likely due to the arousal per se, irrespective of whether it was of a respiratory nature or not [38]. Although it was previously suggested that SB activity might be due to brain stem arousals and not directly from cortical ones [32,34], the cortical arousals are more likely due to an SB event [39]. In fact, a previous study proposed that mastication as a form of activity might lead to cortical activation of different areas [40]. Iida et al. reported in their fMRI study that teeth clenching activated more brain areas than fist clenching, including the bilateral sensorimotor cortex, supplementary motor area, posterior parietal cortex, and dorsolateral prefrontal cortex [41]. It remains unclear whether such activation might result in additional movements during sleep. Boroojerdi et al. reported that voluntary teeth clenching might contribute to movement facilitation in the upper and lower extremities [42]. However, it is yet to be determined whether a similar activity happening involuntarily during sleep would affect motor functions the same way. Furthermore, it is important to note that a model of the protective role of SB in OSA is suggested in previous studies, in which SB events follow apnea/hypopnea, reopening airways and thus ceasing the respiratory event [1,43]. The impact and role of different SB types are also discussed in many studies. For example, reports of Hosoya et al. and Tan et al. have provided evidence for the increased incidence of phasic bruxism events associated with OSA. It can be assumed that at least some part of SB events in OSA patients is evoked in the mechanism of maintaining airway patency [10,44].

Previous studies performed on small samples of SB patients showed that other types of MA, not only limited to temporomandibular joints, might be more frequent in SB patients, but no classification into OSA and non-OSA groups was made in these studies [16,45]. Van der Zaag et al. reported that SB episodes accompanied by PLM events were more common than just “pure” SB events and that combined SB/PLM events with cortical arousal were observed more often than the same SB/PLM events without arousals. The authors concluded that SB, PLM events, and cortical arousals may all share similar neurophysiological mechanisms [35]. Zhang et al. reported that most of the observed SB episodes (especially mixed SB, compared with tonic/phasic phenotypes) were accompanied by PLM events, which usually occur a few seconds prior to SB activity. Thus, they concluded that arousals may lead to a series of motor events more often than just to pure SB activity [36], which is consistent with the findings of the present study. On the other hand, Ohkubo et al. proposed that mastication might be the sole factor leading to cortical activation [40]. It remains unclear whether such activation might further result in MA. Recently, in a small-sample study, Imai et al. reported that there is no specific motor pattern between rhythmic/nonrhythmic jaw movements and bodily movements; however, limb movements are more often observed in SB patients than head/neck and trunk movements. With respect to the lack of specificity of motor patterns, the present study supports this conclusion [46]. However, more studies should focus on the comparison of PLM with other types of body movements, such as MA described in the present study, to determine not only temporal but also causal relationships between different types of movements.

In the non-OSA group, AD and AC were lowest in the NSB subgroup and highest in the SSB subgroup across N1, N2, and REM, but not in N3. In the case of N3 in all subgroups, low activity parameters were observed, which is consistent with those of other studies using other methods of activity measurement [18,20,23,25,30]; in some studies, this feature of N3 is a basic assumption used in sleep scoring systems based on MA [25,30]. Muzet et al. reported that activity in N2 is usually increased before and after REM episodes, but significantly decreased before N3, suggesting that calm sleep without MA might play an important role in shifting to N3 sleep [18]. As previously reported, REM sleep might be significantly longer in SB patients [47]; this change in sleep structure might also contribute to increased MA in N2 in SB patients. SB activity (BEI) in N3 was the lowest among all sleep stages in the present study, which is consistent with the findings of previous studies [47,48,49]. AC and AD in bruxism was comparable to AC and AD in apnea/hypopnea or desaturation in the NSB subgroup, but in the MSB and SSB subgroups and in the whole non-OSA group, the AC and AD in bruxism values were higher than AC and AD in apnea/hypopnea or desaturation, which shows that SB activity could have affected MA more significantly than respiratory episodes alone.

In the OSA group, significant differences in MA across sleep stages in different subgroups were observed only in the case of N1. The NSB, MSB, and SSB subgroups were similar in terms of MA in N2, N3, and REM. Interestingly, BEI values in N1 were much higher than in other sleep stages, which is consistent with previous findings that most of the SB episodes appear during N1 and early N2 sleep [47,49]. The model of the protective function of SB events in OSA, as described earlier, might explain (at least partially) these observations [1,43]. The difference between OSA and non-OSA groups could be because most of the activity could have been related to arousals evoked by respiratory events in the OSA group [38], so no significant differences were observed in the NSB, MSB, and SSB subgroups in terms of other sleep stages; however, this observation needs further research. Nevertheless, in the present study, in the OSA group, AC and AD in desaturation or apnea/hypopnea was higher than AC and AD in bruxism. This observation was valid even in the SSB subgroup when comparing AD in bruxism and AD in desaturation or apnea/hypopnea, but in the case of AC in bruxism and AC in apnea/hypopnea, the values were statistically comparable. AC in bruxism and AC in desaturation were statistically different in the SSB subgroups. These observations could support the hypothesis that in the OSA group, respiratory events could contribute to MA more than SB events.

Regarding SB phenotypes, the strongest correlations with MA parameters in the non-OSA group were observed for phasic bruxism and slightly weaker correlations were observed for mixed bruxism. When combined with nontonic activity, the correlations were stronger than for both the phenotypes separately. However, tonic activity correlated with moderate strength only with AC in bruxism. This suggests that phasic and tonic activities have different origins. These data might support the observation that phasic and mixed activity have an origin similar to those of other sleep movements or that these SB phenotypes might somehow facilitate other movement activities. Michałek-Zrąbkowska et al. reported that snoring was correlated with phasic bruxism regardless of the sleeping position [11], whereas Tan et al. reported that most of the SB episodes that seem to be resulting from respiratory events are also phasic in nature [10].

On the contrary, in the OSA group, only MA in bruxism (referred toas AD and AC in bruxism) was correlated with SB activity—in this case, with all the phenotypes, but the strongest correlations were again observed for phasic bruxism. As for the other MA parameters, no moderate or stronger correlations were observed (there were several significant weak correlations, however).This could suggest that, in apneic patients, MA is rather accompanied by arousals evoked by respiratory events, not SB activity. This observation supports the findings of Kato et al. [38].

Finally, direct comparisons of MA parameters of the OSA and non-OSA group showed interesting observations. AD in bruxism was longer in the whole OSA group and the SSB OSA subgroup, whereas no differences between all subgroups and both main groups were observed in terms of AC. This indicates that, in the SSB OSA subgroup and the whole OSA group, MA episodes last longer, but their general frequency and number are similar to those of the SSB non-OSA group and the whole non-OSA group, respectively. No differences were observed in terms of AC in N1 between all subgroups and main groups, but in the case of AD in N1, differences were observed for the whole groups and the NSB subgroups. Other MA parameters were significantly more pronounced in the OSA groups and the subgroups, probably—once again—because respiratory arousals could contribute to MA more in OSA patients. To the best of our knowledge, no previous studies have directly compared such movement activity parameters between OSA and non-OSA patients focusing on SB activity. In the OSA SSB subgroup, BEI was higher during N1, N2, and REM (compared with the non-OSA SSB subgroup), and not during N3, which could suggest that respiratory events are not likely to increase MA in N3 (as the number of respiratory and SB events in N3 is low, which is consistent with the findings of previous studies [47,49,50,51]). Altogether, these observations would mean that neither SB activity nor respiratory events affect MA in N3. Regarding SB phenotypes, only in the case of SSB subgroups, one comparison was significant—the OSA SSB subgroup had a higher tonic BEI compared with the non-OSA subgroup, which suggests that only in SSB, respiratory events contribute to the increase in tonic activity, which is not observed in more benign types of SB.

Significant differences between groups in terms of age have been presented in this study. OSA participants were older than Non-OSA participants. These observations probably result from the fact that age is a risk factor for OSA [9,52], while it might be a protective factor for SB, as a general tendency of decrease of SB prevalence over age is reported [3]. In this study, there was a weak negative correlation of age, BEI and MA parameters across TST and sleep stages; however, it was more pronounced in the Non-OSA group. At the same time, moderate positive correlations were found in the OSA group for age and AHI along with AD and AC in apnoea/hypopnea and desaturation. This could suggest that age might increase MA related to sleep-disordered breathing, but otherwise it is a rather insignificant factor or a possible weak protective factor for MA. Such a conclusion would be in line with previous studies that reported a general decrease of MA in sleep over age in adults [23,24]. Our findings would also support reports, mentioned in this paragraph, that AHI might increase over age, while BEI is more likely to decrease (at least in Non-OSA group) [3,9,52].

Regarding two main MA parameters measured in this study (i.e., AD and AC in TST), both of these variables were not distributed normally and wide IQR values were measured in some of the subgroups (as presented in Table 1 and Table 4 and on Figure 2 and Figure 3). Broader ranges were observed in OSA groups, especially in NSB and SSB subgroups. The OSA MSB subgroup was less numerous, which could result in a relatively more narrow IQR of AC and AD. Explanation of these wide ranges remains beyond the scope of this study; however, it is now widely discussed that the comorbidity of different sleep disorders is a relatively common phenomenon [53,54,55]. Participants in this study were not checked for comorbid PLM disorder or insomnia, but it was previously reported that such disorders are likely to contribute to increased MA in sleep [29,56]. Similarly, depressive disorders were also reported to affect MA in sleep [28], although, in our study, participants with the history of major psychiatric disorders were excluded. Hypothetically, the reported broad ranges of MA parameters in the OSA group could be explained by comorbid PLM disorder symptoms, insomnia or subclinical depressive symptoms, for which the participants of this study have not been assessed, as it was beyond the scope of the study.

### Strengths and Limitations

This study has a few important limitations. First of all, this is a cross-sectional study with a single observation, so no conclusions based on long-term observations can be made. As the analysis was based on a correlation description of data measured in one time point, it was not possible to determine a clear causal relationship of SB and MA. Due to technical limitations, SB activity and PLM activity were not recorded simultaneously during the same night. This is why the only source of MA data was the sensor located in the central PSG unit. The use of this sensor has other limitations as it is a novel approach to movement monitoring. Thus far, to the best of our knowledge, no other study has recorded MA this way. As the participants were mostly referred to the PSG examination due to the clinical suspicion of SB or OSA, no other phenotypes of jaw activity were scored (such as nonrhythmic masticatory muscle activity).

Although this issue has been discussed, there was a significant difference in terms of age between OSA and Non-OSA participants. A possible explanation of this problem was provided in the Discussion, but the issue still remains a possible limitation of this study. 

Two main MA parameters, AD and AC in TST (shown on Figure 2 and Figure 3), were not distributed normally, and a relatively large IQR of these data was observed. It might be related to the presence of other sleep disorders, which were not a subject of this study, such as PLM syndrome and insomnia, as these disorders could have contributed to the increase of MA. 

This study has some core strengths, among which the large study sample including both OSA and non-OSA groups is important. Even when divided into the NSB, MSB, and SSB subgroups, the sample remained large enough that the statistical analysis of the subgroups was valid. In addition, the sample was properly sex-matched, and age differences were significant only between OSA and Non-OSA participants, while the subgroups within the two main study groups were comparable in terms of age. Furthermore, videoPSG was used as the gold standard for the assessment of OSA and SB, which is also a significant strength, considering the relatively large study sample.

## 5. Conclusions


o
SB is likely to contribute to MA in both OSA and non-OSA groups; however, more research is needed to establish a causal relationship of both phenomena. Further studies, including participants with other sleep disorders (like PLM disorder, insomnia or other, less common, sleep-related movement disorders) are necessary to explore possible mechanisms and relationships underlying different phenotypes of MA.
o
SB events might contribute more to MA than respiratory events in the non-OSA group. In the OSA group, respiratory events might contribute to MA more significantly than SB events.
o
In the non-OSA group, the presence of SB might contribute to an increase in MA in N1, N2, and REM. In the OSA group, this observation is valid only in N1.
o
In the non-OSA group, phasic and mixed SB activity is possibly more related to MA than tonic SB activity. The highest correlation coefficients are observed for phasic and mixed episodes summed up (as nontonic BEI). This difference is probably less in the OSA group.
o
MA in N3 is not likely affected by SB and respiratory events.
o
As age is a known risk factor for OSA, it might contribute to increased MA related to sleep-related breathing disorders, but otherwise it is not likely to affect MA in a significant way.
o
The MA registration method used in this study seems easily applicable, delivers data on different MA parameters and gives an opportunity to register MA in future studies in the field of sleep medicine.

## Figures and Tables

**Figure 1 biomedicines-10-02666-f001:**
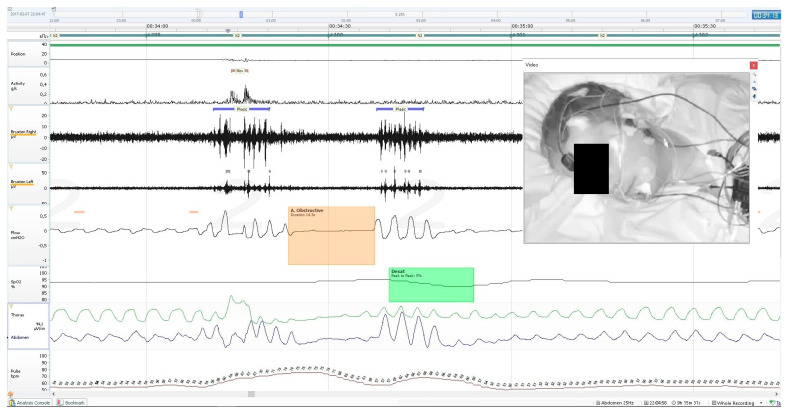
An example of PSG recording, showing a single short MA episode (the uppermost channel), two episodes of phasic bruxism (marked with a purple strip, shown on two widest black canals—registration of the masseter muscle EMG), an episode of obstructive apnoea separating two bruxism episodes (marked with an orange box on the air flow chart) and a desaturation episode following the apnoea (marked with the green box on the pulse oximetry channel). This MA episode occurred within the first phasic bruxism episode, it was a short movement of head, neck and whole body without any change of body position.

**Figure 2 biomedicines-10-02666-f002:**
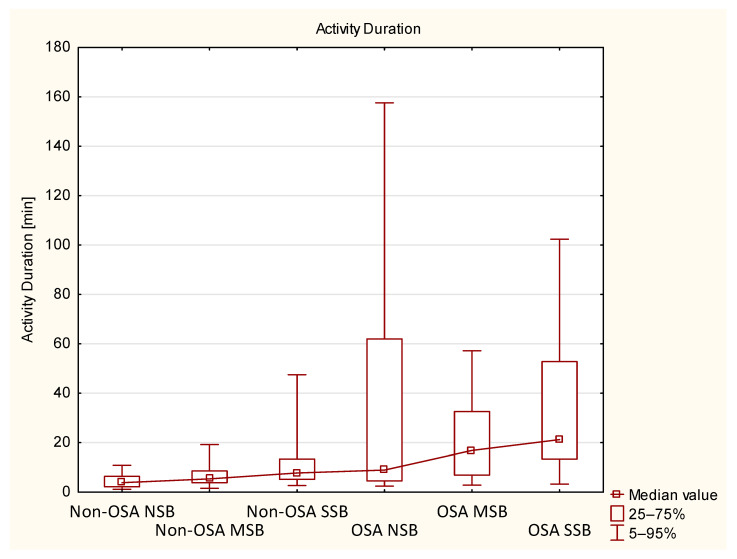
Activity duration in all subgroups presented with median values and percentile ranges. MSB—moderate sleep bruxism, NSB—no sleep bruxism, OSA—obstructive sleep apnea, SSB—severe sleep bruxism.

**Figure 3 biomedicines-10-02666-f003:**
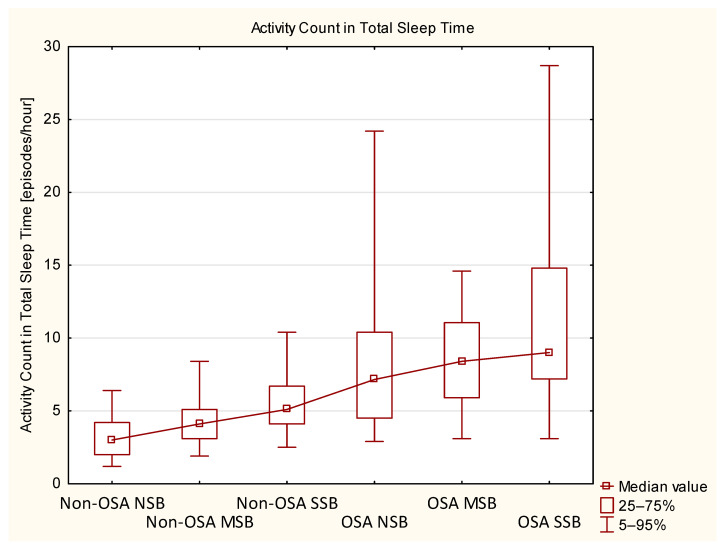
Activity count in total sleep time in all subgroups presented with median values and percentile ranges. MSB—moderate sleep bruxism, NSB—no sleep bruxism, OSA—obstructive sleep apnea, SSB—severe sleep bruxism.

**Table 1 biomedicines-10-02666-t001:** Age and polysomnography parameters in the whole non-OSA group and subgroups. Mean and standard deviation are reported for normally distributed data, while the median and interquartile range are reported for other data. AC—activity count; AD—activity duration; BEI—bruxism episode index; IQR—interquartile range; MSB—moderate sleep bruxism; NSB—no sleep bruxism; SD—standard deviation; SSB—severe sleep bruxism.

	Non-OSA Group (Total)		NSB Group		MSB Group		SSB Group	
	Mean ± SD	Median (IQR)	Mean ± SD	Median (IQR)	Mean ± SD	Median (IQR)	Mean ± SD	Median (IQR)
Age [years]	34.58 ± 10.37	-	37.36 ± 10.75	-	34.69 ± 12.01	-	-	31 (13)
Bruxism episode index (BEI) [episodes/h]	-	3.2 (3.7)	-	1.2 (1.1)	-	2.8 (1.1)	-	6.05 (3.3)
Phasic BEI [episodes/h]	-	1.15 (2.5)	-	0.3 (0.5)	1.18 ± 0.76	-	-	3.75 (3)
Tonic BEI [episodes/h]	-	0.9 (1.1)	-	0.4 (0.7)	-	1.1 (1.1)	-	1.4 (1.9)
Mixed BEI [episodes/h]	-	0.6 (0.7)	-	0.2 (0.4)	-	0.6 (0.4)	-	1 (0.9)
Nontonic BEI [episodes/h]	-	1.9 (3.2)	-	0.5 (0.6)	1.82 ± 0.74	-	-	4.75 (2.7)
Apnea/hypopnea index [episodes/h]	-	1.65 (2.4)	-	1.4 (2.9)	-	1.5 (2.2)	-	2.05 (2.2)
Total sleep time (TST) [min]	434.28 ± 47.72	-	-	442.5 (66)	-	428 (35.9)	437.05 ± 45.54	-
Sleep latency (SL) [min]	-	17.8 (18.3)	-	16.6 (19.7)	-	15.6 (11.8)	-	21.9 (20.4)
REM latency (REML) [min]	-	77.25 (50.2)	-	76 (45.6)	-	75 (52)	-	80.75 (51)
Wake after sleep onset [min]	-	26 (30)	-	27.5 (31)	-	28.3 (28)	-	17.75 (28)
Sleep efficiency [%]	-	88.8 (9.7)	-	88.1 (12.3)	-	87.1 (8.2)	-	90.65 (8.69)
N1% of TST	-	2.6 (2.5)	-	2.6 (2.5)	-	2.2 (2.5)	-	3.2 (3)
N2% of TST	48.97 ± 8.26	-	49.66 ± 7.26	-	49.44 ± 10.54	-	47.98 ± 6.86	-
N3% of TST	24.02 ± 7.3	-	24.59 ± 6.62	-	24.69 ± 7.57	-	-	22.7 (7.7)
REM% of TST	23.54 ± 5.64	-	22.25 ± 4.29	-	22.76 ± 6.91	-	25.32 ± 5.11	-
Arousal index [episodes/h]	-	3.15 (3.3)	-	3.1 (2.3)	-	2.9 (2.5)	-	4.05 (4.3)
BEI in N1 [episodes/h]	-	20.85 (35.9)	-	4.4 (10)	-	21.2 (25.1)	51.71 ± 30.57	-
BEI in N2 [episodes/h]	-	2.1 (3.1)	-	0.6 (0.9)	-	2.1 (1.5)	-	5.15 (4.2)
BEI in N3 [episodes/h]	-	1 (1.9)	-	0.5 (0.9)	-	1.1 (1.7)	-	2.2 (2.4)
BEI in REM [episodes/h]	-	3.05 (4.8)	-	0.7 (2.3)	-	3.4 (3.1)	-	6.1 (5.9)
Activity duration (AD) [min]	-	5.8 (5.8)	-	3.9 (4.2)	-	5.4 (4.7)	-	7.8 (8.3)
AD% of TST	-	1.4 (1.3)	-	0.9 (1.1)	-	1.2 (1.1)	-	1.7 (1.6)
AD in apnea/hypopnea [s]	-	39 (113)	-	26 (67)	-	36 (118)	-	79 (145)
AD in bruxism [s]	-	117.5 (240)	-	24 (43)	-	130 (115)	-	341.5 (372)
AD% in bruxism of total AD [%]	40.33 ± 28.73	-	-	7.14 (18.23)	37.53 ± 20.05	-	-	71.96 (24.84)
AD in desaturation episodes [s]	-	36 (86)	-	26 (67)	-	34 (65)	-	66.5 (141)
AD in N1 [s]	-	54 (89)	-	30 (35)	-	45 (73)	-	108.5 (129)
AD in N2 [s]	-	94.5 (162)	-	72 (142)	-	97 (167)	-	113.5 (189)
AD in N3 [s]	-	31 (53)	-	23 (43)	-	37 (59)	-	35 (55)
AD in REM [s]	-	93 (115)	-	63 (82)	-	97 (113)	-	113 (144)
Activity count (AC) in apnea/hypopnea [episodes/h]	-	2.5 (4)	-	2 (3)	-	2 (3)	-	3 (6)
AC in bruxism [episodes/h]	-	9 (14)	-	3 (3)	9.64 ± 4.19	-	21.93 ± 9.72	
AC in desaturation episodes [episodes/h]	-	2 (3)	-	2 (2)	-	1 (3)	-	3 (4)
AC in N1 [episodes/h]	22.88 ± 16.77	-	-	10 (14.5)	-	22.5 (11.3)	28.62 ± 16.02	-
AC in N2 [episodes/h]	-	2.8 (2.4)	-	2.3 (2)	-	2.7 (3)	-	3.5 (2)
AC in N3 [episodes/h]	-	1.85 (2.1)	-	1.6 (1.5)	-	1.8 (2)	-	2.35 (2.3)
AC in REM [episodes/h]	6.35 ± 4.18	-	-	4.3 (4.9)	-	5.8 (4.9)	-	6.3 (4.5)
AC in TST [episodes/h]	-	4.2 (2.6)	-	3 (2.2)	-	4.1 (2)	-	5.1 (2.5)

**Table 2 biomedicines-10-02666-t002:** Nonparametric *U* test and Kruskal–Wallis ANOVA results in the non-OSA group and subgroups. Significant differences are represented by * and boldface. With the Bonferroni correction applied, all the marked results of K-W ANOVA still remained significant. AC—activity count (activity episodes/h), AD—activity duration (min), MSB—moderate sleep bruxism, NSB—no sleep bruxism, SSB—severe sleep bruxism.

	*p* (Kruskal–Wallis ANOVA)	*p* (NSB vs. MSB)	*p* (MSB vs. SSB)	*p* (NSB vs. SSB)
Activity duration (AD)	**0.0000 ***	**0.011495 ***	**0.012849 ***	**0.000002 ***
AD% of TST	**0.0000 ***	**0.013133 ***	**0.005815 ***	**0.000002 ***
AD in apnea/hypopnea	**0.0016 ***	0.135006	**0.045479 ***	**0.000386 ***
AD in bruxism	**0.0000 ***	**0.000000 ***	**0.000000 ***	**0.000000 ***
AD% in bruxism of total AD	**0.0000 ***	**0.000000 ***	**0.000000 ***	**0.000000 ***
AD in desaturation episodes	**0.0042 ***	0.612787	**0.013575 ***	**0.002067 ***
AD in N1	**0.0000 ***	**0.015576 ***	**0.000338 ***	**0.000000 ***
AD in N2	**0.0116 ***	0.134698	0.161971	**0.002926 ***
AD in N3	0.2129	0.152717	0.846736	0.111030
AD in REM	**0.0007 ***	**0.025478 ***	0.093705	**0.000230 ***
Activity count (AC) in apnea/hypopnea	**0.0050 ***	0.435814	**0.025687 ***	**0.001740 ***
AC IN bruxism	**0.0000 ***	**0.000000 ***	**0.000000 ***	**0.000000 ***
AC in desaturation episodes	**0.0031 ***	0.704895	**0.004106 ***	**0.003805 ***
AC in N1	**0.0000 ***	**0.000144 ***	0.197138	**0.000002 ***
AC in N2	**0.0007 ***	0.074193	0.060053	**0.000126 ***
AC in N3	0.1025	0.189159	0.408728	**0.038061 ***
AC in REM	**0.0013 ***	**0.013269 ***	0.379669	**0.000350 ***
AC in TST	**0.0000 ***	**0.002661 ***	**0.005038 ***	**0.000000 ***

**Table 3 biomedicines-10-02666-t003:** Spearman correlation coefficients calculated for different polysomnography activity parameters and bruxism episode index (total and separately for different bruxism phenotypes) in the non-OSA group. Coefficients > 0.4 (which could be recognized as correlations of at least moderate strength) are represented by boldface. Coefficients that are not statistically significant (*p* > 0.05) are represented by ^+^. AC—activity count (activity episodes/h), AD—activity duration (counted in min), BEI—bruxism episode index.

	BEI	Tonic BEI	Mixed BEI	Phasic BEI	Nontonic BEI
Activity duration (AD)	**0.456**	0.215	0.333	**0.450**	**0.477**
AD% of total sleep time	**0.455**	0.233	0.333	**0.434**	**0.465**
AD in apnea/hypopnea	0.330	0.209	0.181	0.293	0.314
AD in bruxism	**0.829**	0.396	**0.619**	**0.767**	**0.830**
AD in desaturation episodes	0.293	0.125 ^+^	0.168	0.303	0.302
AD in N1	**0.526**	0.378	**0.463**	0.3970	**0.466**
AD in N2	0.237	0.071 ^+^	0.110 ^+^	0.276	0.268
AD in N3	0.142 ^+^	−0.093	0.076 ^+^	0.232	0.220
AD in REM	0.354	0.143 ^+^	0.255	0.377	0.383
Activity count (AC) in apnea/hypopnea	0.272	0.135 ^+^	0.114 ^+^	0.282	0.281
AC in bruxism	**0.876**	**0.401**	**0.657**	**0.823**	**0.882**
AC in desaturation episodes	0.262	0.074 ^+^	0.137 ^+^	0.292	0.288
AC in N1	0.393	0.182	0.311	0.380	0.397
AC in N2	0.330	0.165	0.177	0.329	0.339
AC in N3	0.194	0.002 ^+^	0.090 ^+^	0.278	0.255
AC in REM	0.318	0.140 ^+^	0.201	0.314	0.326
AC in total sleep time	**0.498**	0.244	0.331	**0.487**	**0.506**

**Table 4 biomedicines-10-02666-t004:** Age and polysomnography parameters in the whole OSA group and subgroups. Mean and standard deviation are reported for normally distributed data, while the median and interquartile range are reported for other data. AC—activity count; AD—activity duration; BEI—bruxism episode index; IQR—interquartile range; MSB—moderate sleep bruxism; NSB—no sleep bruxism; SD—standard deviation; SSB—severe sleep bruxism.

	OSA Group (Total)		NSB Group		MSB Group		SSB Group	
	Mean ± SD	Median (IQR)	Mean ± SD	Median (IQR)	Mean ± SD	Median (IQR)	Mean ± SD	Median (IQR)
Age [years]	51.16 ± 13.98	-	56.2 ± 11.36	-	47.58 ± 13.32	-	-	48 (26)
Bruxism episode index (BEI) [episodes/h]	-	3.25 (5.25)	0.9 ± 0.56	-	2.88 ± 0.55	-	-	7.1 (5.6)
Phasic BEI [episodes/h]	-	0.9 (2.8)	-	0.2 (0.4)	-	0.8 (1.3)	-	3.4 (3.5)
Tonic BEI [episodes/h]	-	1 (1.6)	-	0.4 (0.6)	1.05 ± 0.7	-	-	2.2 (2.4)
Mixed BEI [episodes/h]	-	0.6 (0.9)	-	0.1 (0.3)	0.7 ± 0.35	-	-	1 (1.2)
Nontonic BEI [episodes/h]	-	1.8 (3.45)	-	0.4 (0.5)	1.86 ± 0.78	-	-	4.4 (4.2)
Apnea/hypopnea index [episodes/h]	-	22.45 (28.7)	-	30.85 (36.6)	-	11.1 (16.5)	-	21.5 (30.1)
Total sleep time (TST) [min]	411.85 ± 58.52	-	413.48 ± 63.58	-	416.02 ± 52.1	-	-	421.5 (44.9)
Sleep latency (SL) [min]	-	13.55 (19.45)	-	15.6 (20)	23.08 ± 14.37	-	-	10.6 (18.5)
REM latency (REML) [min]	-	81.75 (62.5)	-	82.4 (44)	106.38 ± 59.9	-	-	78.75 (61)
Wake after sleep onset [min]	-	42.25 (45)	-	53.5 (49)	-	38.95 (69.75)	-	37.25 (36.9)
Sleep efficiency [%]	-	84.6 (14.8)	79.62 ± 10.4	-	-	84.2 (16.6)	-	86.45 (10.6)
N1% of TST	-	4.65 (7.3)	-	5.6 (7.8)	-	3.55 (6.5)	-	3.65 (6.9)
N2% of TST	47.83 ± 10.9	-	47.84 ± 9.81	-	49.69 ± 10.38	-	-	48.05 (14)
N3% of TST	23.55 ± 9.67	-	-	23.7 (10.5)	22.44 ± 10.97	-	24.26 ± 10.02	-
REM% of TST	21.58 ± 7.79	-	21.63 ±7.4	-	22.45 ± 8.35	-	21.15 ± 7.98	-
Arousal index [episodes/h]	-	4.75 (7.4)	-	5.05 (6.4)	-	3.15 (6.7)	-	4.85 (8.4)
BEI in N1 [episodes/h]	-	13.6 (34.3)	-	1.5 (6.2)	-	11.6 (8.15)	-	37.3 (27.1)
BEI in N2 [episodes/h]	-	2.8 (5.4)	0.8 ± 0.55	-	2.64 ± 1.06	-	-	7.8 (5.3)
BEI in N3 [episodes/h]	-	1.2 (2.6)	-	0 (1)	-	1.1 (1.9)	-	2.6 (3.5)
BEI in REM [episodes/h]	-	2.1 (3.9)	-	0.6 (1.4)	-	2.45 (2.25)	-	4.5 (3.2)
Activity duration (AD) [min]	-	16.95 (40.5)	-	8.95 (57.4)	-	16.85 (25.65)	-	21.25 (39.4)
AD% of TST	-	4.2 (10.25)	-	2.35 (12.7)	-	4.7 (5.85)	-	4.95 (10.8)
AD in apnea/hypopnea [s]	-	549.5 (1312)	-	284.5 (2935)	-	153 (1176.5)	-	662 (1616)
AD in bruxism [s]	-	196 (496.5)	-	22.5 (76)	-	171 (226)	-	530.5 (676)
AD% in bruxism of total AD [%]	-	20.40 (34.58)	-	4.11 (7.78)	-	19.13 (19.33)	43.17 ± 19.43	-
AD in desaturation episodes [s]	-	629 (1356)	-	310.5 (2787)	-	336.5 (1213.5)	-	730 (1365)
AD in N1 [s]	-	90 (300)	-	64.5 (210)	-	55 (170.5)	-	181 (326)
AD in N2 [s]	-	400 (979)	-	286.5 (992)	-	322 (783.5)	-	534 (913)
AD in N3 [s]	-	93 (527)	-	56 (439)	-	160 (405.5)	-	133 (560)
AD in REM [s]	-	138 (244)	-	106.5 (318)	-	115.5 (130)	-	169 (282)
Activity count (AC) in apnea/hypopnea [episodes/h]	-	20.5 (34.5)	-	20.5 (34)	-	13 (31)	-	23 (43)
AC in bruxism [episodes/h]	-	11 (17.5)	-	3 (3)	9.96 ±4.95	-	-	21 (15)
AC in desaturation episodes [episodes/h]	-	22 (38)	-	22 (36)	-	20 (33.5)	-	26 (43)
AC in N1 [episodes/h]	22.43 ± 14.07	-	-	14.25 (16.6)	20.15 ± 13.02	-	28 ± 13.77	-
AC in N2 [episodes/h]	-	7.8 (6.8)	-	6.45 (7)	-	7.25 (5.6)	-	9.7 (7.1)
AC in N3 [episodes/h]	-	4.5 (5.5)	-	4.2 (6.5)	-	4.5 (5.35)	-	5.1 (3.5)
AC in REM [episodes/h]	-	8.4 (8.2)	-	7.6 (9.2)	8.25 ± 4.65	-	-	9.2 (8.2)
AC in TST [episodes/h]	-	8.5 (5.95)	-	7.15 (5.9)	8.47 ± 4.03	-	-	9 (7.6)

**Table 5 biomedicines-10-02666-t005:** Nonparametric *U* test and Kruskal–Wallis ANOVA results in the OSA group and subgroups. Significant differences are represented by * and boldface. With the Bonferroni correction applied, all the marked results of K-W ANOVA still remained significant with the exception of activity count in TST. AC—activity count (activity episodes/h), AD—activity duration (min), MSB—moderate sleep bruxism, NSB—no sleep bruxism, SSB—severe sleep bruxism.

	*p* (Kruskal–Wallis ANOVA)	*p* (NSB vs. MSB)	*p* (MSB vs. SSB)	*p* (NSB vs. SSB)
Activity duration (AD)	0.0712	0.603279	0.085181	**0.042720 ***
AD% of TST	0.0543	0.611786	0.094340	**0.027581 ***
AD in apnea/hypopnea	0.0946	0.218271	**0.046365 ***	0.197103
AD in bruxism	**0.0000 ***	**0.000087 ***	**0.000019 ***	**0.000000 ***
AD% in bruxism of total AD	**0.0000 ***	**0.000004 ***	**0.000027 ***	**0.000000 ***
AD in desaturation episodes	0.1762	0.548438	0.087240	0.187181
AD in N1	**0.0144 ***	0.406923	**0.007402 ***	**0.036232 ***
AD in N2	0.3296	0.833398	0.406351	0.140665
AD in N3	0.2248	0.257537	0.973679	0.095634
AD in REM	0.2348	0.838221	0.119660	0.181378
Activity count (AC) in apnea/hypopnea	0.1726	0.215734	0.065553	0.442511
AC in bruxism	**0.0000 ***	**0.000000 ***	**0.000000 ***	**0.000000 ***
AC in desaturation episodes	0.3427	0.442766	0.145116	0.457016
AC in N1	**0.0002 ***	0.200170	**0.034706 ***	**0.000041 ***
AC in N2	0.1317	0.738301	0.159161	0.065505
AC in N3	0.5692	0.407014	0.834460	0.352528
AC in REM	0.1527	0.985191	0.130457	0.090857
AC in TST	**0.0253 ***	0.461541	0.094347	**0.010600 ***

**Table 6 biomedicines-10-02666-t006:** Spearman correlation coefficients calculated for different polysomnography activity parameters and bruxism episode index (total and separately for different bruxism phenotypes) in the non-OSA group. Coefficients > 0.4 (which could be recognized as correlations of at least moderate strength) are represented by boldface. Coefficients that are statistically not significant (*p* > 0.05) are denoted by *. AC—activity count (activity episodes/h), AD—activity duration (counted in min), BEI—bruxism episode index.

	BEI	Tonic BEI	Mixed BEI	Phasic BEI	Non-Tonic BEI
Activity duration (AD)	0.196 *	0.146	0.090	0.205 *	0.183 *
AD% of total sleep time	0.209 *	0.174	0.125	0.201 *	0.192 *
AD in apnea/hypopnea	0.124	0.124	0.051	0.156	0.130
AD in bruxism	**0.722 ***	**0.556 ***	**0.526 ***	**0.694**	**0.704 ***
AD in desaturation episodes	0.132	0.116	0.058	0.153	0.127
AD in N1	0.238 *	0.213 *	0.287 *	0.167	0.202 *
AD in N2	0.119	0.117	0.041	0.123	0.092
AD in N3	0.133	0.094	0.026	0.178 *	0.146
AD in REM	0.111	0.047	0.008	0.183 *	0.151
Activity count (AC) in apnea/hypopnea	0.097	0.118	0.044	0.089	0.090
AC in bruxism	**0.909 ***	**0.719 ***	**0.695 ***	**0.809 ***	**0.863 ***
AC in desaturation episodes	0.095	0.118	0.048	0.089	0.093
AC in N1	0.3815 *	0.270 *	0.316 *	0.349 *	0.373 *
AC in N2	0.186 *	0.193 *	0.141	0.160	0.162
AC in N3	0.070	0.088	−0.041	0.063	0.038
AC in REM	0.139	0.147	0.026	0.144	0.130
AC in total sleep time	0.244 *	0.221 *	0.149	0.200 *	0.212 *

**Table 7 biomedicines-10-02666-t007:** Nonparametric *U* test results comparing on-OSA and OSA groups and sleep bruxism subgroups. Significant differences are represented by * and boldface. AC—activity count (activity episodes/h); AD—activity duration (min); BEI—bruxism episode index; MSB—moderate sleep bruxism; NSB—no sleep bruxism; SSB—severe sleep bruxism.

	*p* (Non-OSA vs. OSA)	*p* (Non-OSA NSB Group vs. OSA NSB Group)	*p* (Non-OSA MSB Group vs. OSA MSB Group)	*p* (Non-OSA SSB Group vs.OSA SSB Group)
Age	**0.000000 ***	**0.000000 ***	**0.000270 ***	**0.000000 ***
Bruxism episode index (BEI)	0.734328	0.168337	0.914507	0.089253
Phasic BEI	0.687735	0.143144	0.681880	0.696352
Tonic BEI	0.125748	0.828866	0.884567	**0.001062 ***
Mixed BEI	0.756226	0.209929	0.284976	0.960735
Nontonic BEI	0.668843	0.109258	0.939683	0.667070
BEI in N1	**0.023683 ***	0.163457	**0.009948 ***	**0.013838 ***
BEI in N2	0.060115	0.652119	0.053735	**0.000856 ***
BEI in N3	0.888649	**0.041670 ***	0.899257	0.114499
BEI in REM	**0.028534 ***	0.212045	0.177434	**0.030822 ***
Activity duration (AD)	**0.000000 ***	**0.000009 ***	**0.000418 ***	**0.000000 ***
AD% of TST	**0.000000 ***	**0.000002 ***	**0.000092 ***	**0.000000 ***
AD in apnea/hypopnea	**0.000000 ***	**0.000000 ***	**0.000033 ***	**0.000000 ***
AD in bruxism	**0.011545 ***	0.382733	0.185866	**0.003281 ***
AD% in bruxism of total AD	**0.000043 ***	**0.020023 ***	**0.001982 ***	**0.000000 ***
AD in desaturation episodes	**0.000000 ***	**0.000000 ***	**0.000005 ***	**0.000000 ***
AD in N1	**0.000272 ***	**0.000140 ***	0.536917	0.111372
AD in N2	**0.000000 ***	**0.000001 ***	**0.000265 ***	**0.000001 ***
AD in N3	**0.000000 ***	**0.007223 ***	**0.001463 ***	**0.000036 ***
AD in REM	**0.002472 ***	**0.021359 ***	0.360942	0.123027
Activity count (AC) in apnea/hypopnea	**0.000000 ***	**0.000000 ***	**0.000000 ***	**0.000000 ***
AC in bruxism	0.563350	0.993801	0.884428	0.483147
AC in desaturation episodes	**0.000000 ***	**0.000000 ***	**0.000000 ***	**0.000000 ***
AC in N1	0.889271	0.081405	0.226299	0.930479
AC in N2	**0.000000 ***	**0.000000 ***	**0.000065 ***	**0.000000 ***
AC in N3	**0.000000 ***	**0.000061 ***	**0.000406 ***	**0.000005 ***
AC in REM	**0.000010 ***	**0.001090 ***	0.192193	**0.004807 ***
AC in TST	**0.000000 ***	**0.000000 ***	**0.000013 ***	**0.000000 ***

**Table 8 biomedicines-10-02666-t008:** Spearman correlation coefficients calculated for different polysomnography activity parameters and age in all groups and subgroups. Coefficients > 0.4 (which could be recognized as correlations of at least moderate strength) are represented by boldface. Coefficients that are statistically significant (*p* < 0.05) are denoted by *. AC—activity count (activity episodes/h), AD—activity duration (counted in min), AHI—Apnea/Hypopnea Index, BEI—bruxism episode index, MSB—moderate sleep bruxism; NSB—no sleep bruxism; SSB—severe sleep bruxism.

	Non-OSA	Non-OSA NSB	Non-OSA MSB	Non-OSA SSB	OSA	OSA NSB	OSA MSB	OSA SSB
BEI	−0.177 *	−0.234	0.205	0.024	−0.242 *	−0.006	−0.297	−0.032
Phasic BEI	−0.150	−0.266	0.162	−0.008	−0.220 *	−0.006	−0.091	−0.049
Tonic BEI	−0.130	−0.179	−0.054	0.141	−0.162	0.025	−0.181	0.101
Mixed BEI	−0.119	−0.035	0.101	−0.103	−0.196 *	0.068	0.045	0.015
Non-Tonic BEI	−0.152	−0.152	0.155	−0.029	−0.253 *	−0.048	−0.028	−0.115
AHI	0.169 *	0.292 *	0.222	0.052	**0.464 ***	0.124	**0.629 ***	**0.530 ***
Activity duration (AD)	−0.234 *	−0.132	−0.165	−0.271 *	0.057	0.024	0.276	0.098
AD% of total sleep time	−0.240 *	−0.087	−0.186	−0.309 *	0.058	0.028	0.270	0.087
AD in apnea/hypopnea	−0.134	0.035	−0.148	−0.164	0.230 *	0.066	**0.423 ***	0.315 *
AD in bruxism	−0.205 *	−0.100	0.058	0.125	−0.176	0.006	0.003	−0.011
AD in desaturation episodes	−0.036	0.044	0.057	−0.079	0.213 *	0.072	**0.435 ***	0.279 *
AD in N1	−0.119	−0.301 *	0.042	0.088	0.120	0.015	0.166	0.276
AD in N2	−0.205 *	−0.066	−0.124	−0.342 *	0.186 *	0.085	0.399	0.238
AD in N3	−0.236 *	−0.172	−0.179	−0.307 *	0.024	0.127	0.081	0.019
AD in REM	−0.253 *	−0.192	−0.169	−0.319 *	−0.137	−0.146	0.011	−0.175
Activity count (AC) in apnea/hypopnea	−0.091	0.046	−0.098	−0.109	0.287 *	0.049	**0.464 ***	0.383 *
AC in bruxism	−0.185 *	−0.105	0.038	−0.142	−0.240 *	−0.007	−0.236	−0.026
AC in desaturation episodes	0.034	0.074	0.103	−0.049	0.325 *	0.101	**0.517 ***	**0.409 ***
AC in N1	−0.102	−0.305 *	0.092	0.051	−0.212 *	−0.057	0.034	−0.340 *
AC in N2	−0.214 *	−0.061	−0.111	−0.313 *	0.232 *	0.099	**0.529 ***	0.288 *
AC in N3	−0.205 *	−0.126	−0.084	−0.314 *	0.062	0.050	−0.068	0.103
AC in REM	−0.232 *	−0.216	−0.218	−0.231	−0.110	−0.193	−0.040	−0.053
AC in total sleep time	−0.253 *	−0.177	−0.183	−0.265	0.090	−0.015	0.341	0.185

## Data Availability

Not applicable.

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
