# Peer review of "Sleep Bruxism Contributes to Motor Activity Increase during Sleep in Apneic and Nonapneic Patients—A Polysomnographic Study"

_biomedicines, 2022, doi:10.3390/biomedicines10102666_

Round 1

Reviewer 1 Report

Dear Authors, I congratulate you for a well written article.

Author Response

Dear Reviewer, thank you very much for this compliment and your support.

Reviewer 2 Report

Dear Authors the paper ”Sleep bruxism contributes to motor activity increase during  

sleep in apneic and nonapneic patients - a polysomnographic  study. ‘’is really interesting, well conducted and fits the objectives of the journal; but it is necessary to review some points in order to improve the quality of the paper: 
-First, i ask you to check the plagiarism of your article using specific sites to get a similitary report

-Please be sure to use only keywords accordingly to medical subject headings (Mesh word) for a better indexing.

- The introduction section is very short and is needed to add other references to increase the quality of the manuscript, Add recent references about the topic of the article, dwelling in the introduction on articles published in 2022 and describing what your article will add compared to the last articles published; Preferably a published articles should be with 90 or more references.

I suggest you some articles that will help you improve your article.

Efficacy of conservative approaches on pain relief in patients with temporomandibular joint disorders: a systematic review with network meta-analysis. PMID: 36148997.

Teledentistry in the Management of Patients with Dental and Temporomandibular Disorders Doi: https://doi.org/10.1155/2022/7091153

Stem Cells in Temporomandibular Joint Engineering: State of Art and Future Persectives. The Journal of Craniofacial Surgery: October 2022 - Volume 33 - Issue 7 - p 2181-2187 doi: 10.1097/SCS.0000000000008771

-You need to review the grammar and English of your article.

-I suggest you to add an image in order to improve the iconography of the article.

-I suggest you add a table with the list of abbreviations used in the text.

-Please expand conclusion section with main results and future perspectives of this study

Thank You,

Kind Regards

Author Response

Dear Reviewer, thank you very much for all the effort and support you have given to improve the quality of our paper. We have considered all your suggestions and concerns, a short description of our actions may be found in the attached file.

Reviewer 3 Report

Tables 1,4, The table is too busy and with data that does not contain the information required for statistical evaluation. It is not clear why there are not two tables. One for normally distributed variables with its mean and standard error (such as age) and a second table for non-normally distributed variables with median and interquartile range. The attempt to include everything in one table creates clutter and confusion and makes it difficult to gain insights from it.

In the sample according to row 189 age=42.15±14.69. In Table 1 regarding the non-OSA group, all the average values are under the age of 37.36. That is, the other groups are older. How does this affect the findings? How was this difference neutralized in the statistical analyses?

The multiple comparisons in Table 2 did not indicate whether a required Bonferroni correction was performed.

In the correlations in table 3, 6, 7 precisions above 4 decimal digits is unnecessary and does not contribute to the understanding of the findings. must be reduced.

Figure 1. The distributions are Poisson because they start at zero on the timeline. Poisson analysis may have been superior to a-parametric analysis. It must be explained why the margin of activity is so wide in the OSA NSB group.

Figure 2. Certainly, a Poisson distribution from the very essence of the measurement of counting activities. Why are the OSA NSB and OSA SSB groups so wide?

שמירת התרגום

It is not clear how it was concluded that SB is the cause of MA and not the other way around. A correlation was found but not a causal relationship. All causal terms should be carefully checked in conclusions.

Author Response

Dear Reviewer, thank you very much for all the effort and support you have given in order to improve the quality of our paper. We have considered all your suggestions and concerns and a short description of our actions may be found in the attachment.
